# Interpersonal Communication in Intensive Care Units: A Qualitative Study on Family Members’ Experiences in a Turkish Public Hospital

**DOI:** 10.3390/healthcare13233100

**Published:** 2025-11-28

**Authors:** Asu Ozgultekin, Elgiz Yilmaz Altuntas, Deniz Birtan

**Affiliations:** 1Department of Anaesthesiology and Resuscitation Intensive Care, Martyr Prof. Dr. İlhan Varank Education and Research Hospital, Health Sciences University, 34785 Istanbul, Turkey; 2Faculty of Communication, Galatasaray University, 34349 Istanbul, Turkey; elyilmaz@gsu.edu.tr; 3Organ Transplantation Unit, Pendik Training and Research Hospital, Marmara University, 34899 Istanbul, Turkey; denizbirtan@hotmail.com

**Keywords:** ICU intensive care units, health communication, healthcare professionals-family members engagement, interpersonal communication

## Abstract

**Highlights:**

**What are the main findings?**
Face-to-face communication between healthcare professionals and patients’ relatives in adult ICUs is essential for fostering trust, emotional support, and collaborative decision-making, thereby mitigating anxiety and uncertainty in critical care contexts, even under the conditions of a state hospital with limited resources.Face-to-face interactions improve the accuracy, clarity, and comprehension of complex medical information, enhancing relatives’ involvement in care processes and satisfaction with communication quality in intensive care units.

**What are the implication of the main findings?**
Healthcare systems and ICU teams should prioritize structured, regular face-to-face communication strategies to strengthen family engagement, promote shared decision-making, and improve overall quality of care in critical settings.Training and institutional policies that support effective in-person dialogue with patients’ relatives may reduce psychological distress, enhance satisfaction with care, and foster more ethically sound and patient-centered intensive care experiences.

**Abstract:**

**Background/Objectives**: Studies on the satisfaction of patients’ relatives in intensive care units are quite limited both in our country and worldwide. In intensive care units, particularly in adult settings, communication is known to be one of the most important factors influencing patient and family satisfaction. From a communication theory perspective, there are very few qualitative descriptive studies that reveal how this issue is perceived. This research aims to examine the information needs of relatives of patients receiving treatment in intensive care units and their satisfaction levels with regard to regular information provision practices carried out by healthcare professionals working in intensive care units in Turkey. **Methods**: Semi-structured interviews were conducted with 23 patients’ family members in two adult ICUs at a university-affiliated training and research hospital in Turkey. In the data collection process, the ‘Critical Care Family Needs Inventory’ was used to establish the family needs, as well as a sociodemographic questionnaire that included: age, gender, educational level, patient relationship and previous ICU experience. The unstructured texts obtained from the interviews were analyzed using the Atlas.ti qualitative data analysis software for the thematic analysis method. **Results**: The findings revealed that while face-to-face information provided by healthcare professionals is generally perceived as comprehensive, regular, and confidence-building, the experience of obtaining information by telephone varies greatly depending on hospital and family circumstances. Inconsistencies in telephone-based information access can create significant communication barriers for some families, yet in certain situations (e.g., chronic illnesses), it can serve as a vital adaptation and information flow tool. **Conclusions**: The ‘Uncertainty Management Theory’ and the ‘Information Management Theory’ are critical for understanding the effects of communication quality in the intensive care unit (ICU) environment on the psychological state of family members and their decision-making processes. Healthcare professionals should recognize that their communication serves not only an informative function but also has profound effects on family members’ psychological well-being and participation in the healthcare process.

## 1. Introduction

Intensive Care Units (ICUs) are specialized hospital departments where patients in critical condition or at terminal stages are admitted [1]. These units involve numerous invasive and non-invasive procedures [2], as well as treatment and care practices [3,4]. Admission of critically ill patients to the ICU [5] and visiting them during their stay [6] are stressful experiences not only for patients and healthcare professionals [7] but also for patients’ families [3,5]. In ICUs, treatment decisions often need to be made rapidly, usually under conditions of limited or uncertain diagnostic information. During the care of patients, meeting the needs of their families [8,9] and providing support [4] are also considered fundamental responsibilities of the ICU team. The unfamiliar environment in which family members find themselves [10] disrupts family functioning and integrity, leading to negative psychosocial outcomes when a member is diagnosed with an acute or chronic illness [5]. Sources of family dissatisfaction include inconsistent information, unclear roles in the decision-making process, and inadequate understanding of diagnosis, prognosis, or treatment [11]. The communication style and information-sharing practices of the ICU team play a decisive role in family satisfaction.

As Gürkan [12] emphasized, Molter was the first to investigate the needs of ICU patients’ families in 1979. His study revealed that the fundamental needs of family members include obtaining information about their loved one’s condition and being able to remain close to them. In ICUs, patients are often either in critical condition or sedated, which prevents them from expressing their treatment preferences and values [13]. In such cases, family members act as surrogate decision-makers [3,13]. As Frivold et al. [9] pointed out, the Patient Rights Charter stipulates that, in accordance with the principle of patient autonomy, medical interventions of a serious nature must be based on informed consent. The process in which families and healthcare professionals participate equally is referred to as shared decision-making. To be effective, family members must be sufficiently informed to weigh potential benefits and risks of treatment and to understand possible outcomes for the patient [9].

During the ICU stay, it is crucial for family members to receive regular updates and to build trust in the ICU team in order to manage the psychological burden. In Henrich’s [14] study, family satisfaction with ICUs was evaluated in terms of care, decision-making, and communication. Providing information about visiting schedules, enabling communication with the treating physician, fostering trust among family members, and offering information both by telephone and in person, when possible, were identified as important factors that enhance family satisfaction.

Families of patients who are unexpectedly admitted to the ICU face an urgent need for effective communication during the sudden transfer process and throughout subsequent stages. It is essential to explain the sudden situation clearly to relatives, to take their needs and expectations seriously, and to ensure they feel emotionally supported [15]. Organizational support provided to ICU patients’ relatives can reduce their psychological stress and enable them to offer more consistent and effective support to patients [16]. Patient and family satisfaction is widely recognized as a key indicator of quality of care in ICUs [17].

Although ineffective communication is known to negatively influence patient and family satisfaction in ICU settings—particularly in adult ICUs—qualitative research analyzing communication-related problems from the perspective of communication theory remains limited [3,18,19,20,21,22,23,24,25]. This study was conducted to examine the impact of communication quality between healthcare professionals and patients’ families through the lens of information management and communication framing theories in Turkey. Since healthcare professionals are the primary source of information for families in ICU, this study focuses initially on family members’ perspectives while recognizing the importance of their hospital experiences.

## 2. Materials and Methods

### 2.1. Study Design and Research Settings

This research adopts a constructivist/interpretivist epistemological stance, grounded in the assumption that knowledge is constructed through individual experiences and social interactions rather than being objectively discovered. According to the interpretivist paradigm, social reality is not fixed; instead, it is socially constructed and sustained by individuals’ interpretations [26]. The study aims to analyze and understand the subjective experiences and interpretations of intensive care patients’ relatives regarding the information-sharing process.

The settings were two units of level three (severely ill ICU patients were admitted) mixed ICUs, each consisted of 22 beds in a university-affiliated training and research hospital in Istanbul. These units provide the highest level of intensive care: that is complex, multisystem life support for critically ill patients including invasive and noninvasive mechanical ventilation, various oxygen therapies, continuous renal replacement therapies, invasive and noninvasive monitorization of all systems.

The patients with multiple organ failure (e.g., respiratory, renal, cardiovascular), postoperative or trauma cases with unstable physiology, septic shock, ARDS (Acute respiratory distress syndrome), or patients with severe intracranial pathologies, or in coma state are all managed in these units. The overall mortality rates are usually between 300 and 50%. The nurse-to-patient ratio was generally 1 to 2. The occupancy rate of the ICUs was 95–110% (when a patient was discharged, another one was admitted on the same day). The mean length of stay was 7.5 days. The mean age of the patients was 76.6 years. Two consultant intensivists and 4–5 resident doctors in each ICU were in charge of supervising the care of the patients. Different specialities were called for consultations to take part in the treatment of the patients.

### 2.2. Study Population

Patients’ diagnoses included all medical and surgical life-threatening conditions like septic shock, pneumonia, multisystem organ failure (heart, lung, liver and kidney), gastrointestinal hemorrhage, intracranial bleeding-stroke, postoperative complications, trauma cases, as well as malignancies with systemic problems. Typical critical care interventions included invasive and noninvasive mechanical ventilation, various oxygen therapies, invasive and noninvasive monitorization of all systems.

In this study, our sample consisted of family members aged 18–85 whose patients had been hospitalized in the ICU for at least 72 h. This selection criterion was used to create a homogeneous group sharing similar experiences with the information-sharing process, which directly relates to our research question: “*What have been your experiences regarding the information practices provided by healthcare professionals since your patient was admitted to the Intensive Care Unit (ICU)?*” Therefore, we categorize our sampling strategy as homogeneous purposive sampling. Homogeneity ensures that participants’ experiences are comparable, which is essential for providing a clear answer to the research question [27,28].

The primary reasons for ICU admission of the selected patients were pneumonia (n = 5), respiratory insufficiency due to chronic obstructive lung disease (n = 4), sepsis (n = 3), postoperative hemodynamic-respiratory problems (n = 3), gastrointestinal hemorrhage (n = 1), intracranial bleeding-stroke (n = 2), acute heart failure (n = 1), renal failure (n = 2), acute pancreatitis (n = 1), diabetic coma (n = 1). The patients under the severe risk of death were excluded from the study (in shock state, terminal malignancy, vegetative state, etc.)

Participants consisted of 23 relatives of these patients receiving treatment in the intensive care unit and having high probability of survival. Patients with a higher life expectancy were selected for the study to enable their family members to better evaluate the communication service. There was no difference between the families of patients in ICU1 and ICU 2, in terms of education, socioeconomic status, or prior experiences.

Eligible participants were family members of patients, aged between 18 and 85, who had stayed in the Intensive Care Unit (ICU) for a sufficient period (at least 72 h) to experience the information-sharing process. These criteria were established in line with the aims of the qualitative study, specifically to collect data from individuals with a similar level of critical care experience. In accordance with the defined criteria, suitable relatives of patients were invited to participate in the study either in person or via telephone during the designated ICU information hours. Individuals who agreed to participate were given detailed verbal information by the researchers prior to the interview, and their consent was obtained. Written permission was additionally obtained from participants for the interviews to be recorded using a voice recorder; the recordings were deleted after the transcription process was completed. Researchers (EYA, DB) took field notes during the interviews. Participants were informed that participation in the research was entirely voluntary and that they could withdraw from the project at any time. The participants in the study consisted of 23 relatives of patients receiving treatment in the intensive care unit. Information was obtained from 12 of these relatives in person and 11 by telephone. According to the protocol, the way of the information that was given to the relatives on the daily basis and the permission of visiting the patient by the relative were designed in two different forms and those two protocols were started to be performed two weeks before the study was begun. The medical director of the ICU, senior resident doctors, a communication specialist and a specialist in the history of medicine and ethics were carried out the study.

After two rounds of training, the first interviewer* coded all the interviews in the sample (n = 23) and the second interviewer* coded a randomly selected 20% of the articles (n = 5). Cohen’s kappa was calculated to assess intercoder reliability for each coding category: sociodemographic questionnaire (κ = 1.00), face to face information gathering experiences (κ = 0.94), experiences of obtaining information by telephone (κ = 0.92), communication competence of health professionals (κ = 0.93), adequacy of information (κ = 0.91) and previous hospital experiences (κ = 1.00).

In the ICU-1, the family members were informed about the patients only by daily telephone calls and the visiting the patient at the bedside was strictly limited.

In the ICU-2, the family members were invited to the hospital to be informed about the patients treatment process and they were allowed to see the patients at the bedside whenever they want. The information was given by a phone call only if they could not come or prefer not to come to the hospital. In this context, semi-structured in-depth interviews were conducted.

* The interviews and data analysis in this study were conducted by an ethics and by a health communication specialist/researchers with over three years of professional experience in qualitative research methods. One of the interviewers is an expert in ethics-based data collection procedures and the protection of participant rights. And the second interviewer is an expert on interpersonal health communication, especially physician-patient communication. There was no prior personal, professional, or academic relationship between the participants and the interviewers.

### 2.3. Data Collection

The research data was collected between November–December 2024. Participation was completely voluntary and the consent of the participants was obtained through the informed consent text at the beginning of the questionnaire form. In the data collection process, the ‘Critical Care Family Needs Inventory’ [29] was used to establish the family needs, as well as a sociodemographic questionnaire that included: age, gender, educational level, patient relationship and previous ICU experience.

A homogeneous sampling method was used to identify the differences and similarities between the experiences of the participants. Interviews were concluded after data saturation was achieved, which is the point in qualitative research where obtained information begins to repeat and no new code, category, or theme emerges [28]. In this study, saturation was determined by the criterion that no new thematic element emerged during the analysis of data collected from the 23 relatives of patients. These interviews were conducted both in person and by telephone. Participants who agreed to participate in the study were informed in detail and comprehensively about the study by the researchers. All interviews were recorded using a voice recorder. The flow of questions during the interviews was flexible and did not follow the order specified in the interview protocol. All interviews were conducted in accordance with confidentiality rules with the consent of the participants, and field notes were taken by the researchers. The socio-demographic data of the participants were analyzed using the SPSS-27 programme in terms of frequency and percentage distributions. Please see Appendix A for details. The unstructured texts obtained from the interviews were analyzed using the Atlas.ti-21 qualitative data analysis software. Thematic analysis, a qualitative data analysis technique [30], was implemented, establishing in codes, and categories according to Saldana [31].

Audio files were deleted once the transcription process was complete. Transcripts were stored on a secure and encrypted server accessible only to the research team. All transcripts and excerpts were anonymized to ensure that participants’ identities could not be revealed, and personal information obtained during the interviews was protected. These procedures ensured data security in compliance with both ethical and legal requirements.

Participants were identified (coded) according to the number assigned to them in the study, their relationship to the patient, gender, and the communication channel through which they received information (telephone or face-to-face). For example, 1.S.M.T (Participant 1, Son, Male, Telephone), 2.D.F.T (Participant 2, Daughter, Female, Telephone) 3.S.M.F (Participant 3, Son, Male, Face-to-face), 4.D.F.F (Participant 4, Daughter, Female, Face-to-face)

## 3. Results

Descriptive statistics and frequency distributions of the variables used in the analysis, describing the current situation, are summarized in Table 1 above.


*Thematic Analyses*


Consistent with the objectives of this research, five major categories emerged and were categorised into themes (Table 2). This coding scheme was developed based on the topic categories in existing studies [32,33,34,35,36].

The interviews revealed important findings regarding the differences between patients’ relatives’ experiences of obtaining information from healthcare professionals in person and by telephone, and the effectiveness of these channels. The analyses showed that face-to-face communication (Table 3) is generally the primary and most comprehensive source of information, while access to information by telephone (Table 4) is inconsistent between interviews and leads to different experiences. As highlighted by the findings of our study, the content, frequency and form of information provided to patients’ relatives significantly contributes to their emotional coping abilities, their ability to make informed decisions and to develop appropriate expectations.

While face-to-face information provided by healthcare professionals is generally perceived as comprehensive, regular, and confidence-building, the experience of obtaining information by telephone varies greatly depending on hospital and family circumstances. Inconsistencies in telephone-based information access can create significant communication barriers for some families (e.g., “*We want them to speak openly to us. When those providing information are nervous, I become nervous too, and this nervousness irritates me. Since we can’t see them constantly, I do have some doubts inside me*”), yet in certain situations (e.g., chronic illnesses), it can serve as a vital adaptation and information flow tool.

These findings demonstrate that communication in intensive care is not merely an activity of medical information transfer, but also a management process that reduces uncertainty and focuses on the relationship. Effective information management (clear, consistent, face-to-face, and tailored to the recipient) enhances trust by enabling families to better manage uncertainty surrounding the critical condition, thereby reducing negative emotional effects (anxiety).

Just like a map, the information provided by healthcare professionals serves as a guide for patients’ relatives navigating the complex and foggy terrain of the intensive care journey. The more detailed and clear the map is (Clear Instructions, Building Trust), the less likely the traveller is to get lost (Reduced Uncertainty) and the more confidently they continue their journey (Psychological Relief).

Analyses of healthcare professionals’ communication skills reveal (Table 5) that, in general, they demonstrate strong competence in this area. This competence forms the basis of knowledge adequacy and significantly influences families’ experiences of healthcare services.

However, some limitations or differences in communication skills have also been observed:-Communication Gaps: There are situations where family members have difficulty obtaining information by telephone.

Our findings reveal that face-to-face interactions play a vital role in strengthening families’ trust in healthcare professionals. Consistent information flow can maintain trust, even when the news is negative. This reduces cognitive uncertainty; families feel more secure knowing they are receiving accurate information about the patient’s condition, no matter how dire it may be.

-Inconsistent Information Flow: Despite daily updates being provided, sometimes this information is only given when family members visit the hospital in person.-Unexplained Terminology: Some medical terms are inadequately explained, which may initially cause confusion for family members (e.g., “*We are currently feeding him using a separate device called a nasogastric tube, which goes from his nose to his stomach. In other words, we are feeding him by sending food directly from his nose to his stomach*”).

Overall, the findings emphasize that communication competence combines technical proficiency in conveying medical details with interpersonal skills that enable effective and supportive interactions with families in critical care settings. This competence contributes significantly to family satisfaction and trust.

The analyses presented under the heading ‘Information Adequacy’ (Table 6) in the texts reveal that the information provided by healthcare professionals is generally available and important, but varies in terms of consistency and scope. This directly affects families’ ability to understand and manage the treatment process.

However, the content of the news received, i.e., whether the news is positive or negative, directly affects the family’s emotional response. Family members have stated that they appreciate the doctors’ candour and honesty. Information supports the formation and maintenance of trust, even if it is negative. We put forward that the information can change family members’ decision-making processes and preconceptions. For example, negative preconceptions about intensive care can be overcome through direct experience and ongoing information sharing.

Although families’ previous hospital experiences are generally addressed to a limited extent in healthcare provision, it has been observed that these experiences (Table 7) have a significant impact on the perception of the current situation and communication. In some texts, this issue has been highlighted, particularly in terms of preconceptions about intensive care processes.

Previous experiences shape the need for education and openness to education. Negative previous experiences can create trust issues that affect the perception of current information and communication. Previous experiences provide a context that shapes both the healthcare professional’s communication strategy and the family’s interpretation of information. They can also influence the family’s expectations and questions regarding current procedures.

We may suggest regarding our findings that the fact that a family with negative preconceptions about intensive care reported a change in their perceptions thanks to a constant flow of information and face-to-face experiences demonstrates that information helps overcome uncertainty by reorganizing the cognitive structure related to the situation.

## 4. Discussion

The interview transcripts reveal significant effects of physician-provided information on patients’ relatives, demonstrating how clinical communication influences family members’ psychological states, decision-making processes, and overall healthcare experience.

The transcripts demonstrate clear emotional consequences resulting from physician-provided information: For example, in an interview mentioned above, when explicitly asked about emotional responses to information, the patient’s sister states: “ e.g., *Well, because we don’t receive good news, we don’t feel good. Of course, if you received good news, you would be happier; since I don’t receive good news, I don’t feel good*” This statement directly connects information content with emotional well-being. Similarly, in an another interview, when asked about anxiety levels, the family member acknowledges: “e.g., *I have anxiety one day, I hope the next, I don’t know*”. This fluctuation in emotional state appears connected to the variable prognosis information they receive.

Physician-provided information significantly influences trust development. For example, one of the relatives repeatedly emphasizes trust despite receiving unfavourable news: “e.g., *We trust the doctors, you know*”. This suggests that consistent information provision, even when negative, can maintain trust in the healthcare relationship. In analyzed interview contents, the healthcare provider’s transparency about treatment modifications and clear explanations of medical interventions (e.g.*, “We were giving pressurised oxygen therapy, as you know, but we plan to reduce the number of patients and discharge them from the ward.”* appears to foster a collaborative atmosphere where family members feel informed and involved. Thus, we may suggest that compared to the Western settings, communication tends to be more hierarchical model (senior physicians having greater authority in decision-making) in Turkish ICU culture. Family involvement is expected, but physicians tend to be filter information to protect families or avoid distress. Communication is often indirect and protective; physicians tend to soften bad news. Emotional support is often given through personal warmth and empathy rather than structured communication frameworks.

We put forward also the impact of the use of medical terminology by healthcare professionals. Based on the available contents’ evidence, the rate of medical jargon appears moderate rather than high. The terminology used generally relates to common clinical parameters (blood pressure, infection, oxygen) rather than highly specialized medical concepts. While some medical terms are used without explicit explanation, there is no clear evidence that this impeded family comprehension or satisfaction with communication. This finding suggests that physicians in these cases may have calibrated their language appropriately to the family’s understanding, particularly given the ongoing nature of these healthcare relationships.

These findings align with communication frameworks in healthcare that emphasize the bidirectional relationship between information provision and recipient response. Specifically:*Uncertainty Management Theory:* The transcripts demonstrate how physician information helps family members navigate the inherent uncertainty of critical illness (e.g., “*I come every day, and every day I receive information face to face and go inside, God bless them. They let me in because the patient is doing a little better, thank goodness. We received good news today too”*.). Information, even when negative, appears to provide cognitive structure that helps relatives process the situation. In the context of clear instructions, the uncertainty in an intensive care setting relates not only to diagnosis but also to practical and procedural matters (When can I visit? Who should I contact? What is the next step?). In addition to this, the fluctuation in emotional state (anxious one day, hopeful the next) being linked to the variable forecast information they receive confirms that the content of the information directly affects emotional well-being and that emotional distress persists when uncertainty is not completely eliminated.*Relationship-Centred Care:* The emphasis on regular information provision (e.g., “*We get information every day”*) reflects an approach that recognizes family members as integral to the care ecosystem. Within the context of Information Management Theory, when healthcare professionals demonstrate transparency regarding treatment changes and complex interventions, families feel informed and involved in the process.*Health Belief Model:* Information from physicians appears to influence family members’ perceptions of situation severity and potential outcomes (e.g., “*We admitted your mother due to pneumonia. As we already informed you, we started antibiotic treatment on the day she was admitted. The infection parameters in her blood have clinically stabilised as of today”*), which in turn affects their emotional responses and decision-making.

In addition to this, our current findings are fully consistent with the key factors affecting family satisfaction and quality of care, as highlighted in studies such as Haave et al. [3], Ferrando et al. [21], and Stricker et al. [25]. This consistency reinforces the universal importance of communication in ICU success. Our study demonstrates that face-to-face communication between healthcare professionals and patients’ relatives is fundamental in strengthening trust, emotional support, and collaborative decision-making. This finding aligns with the key factors highlighted by Stricker et al. [25] that enhance family satisfaction, such as providing opportunities for communication with doctors, fostering trust, and ensuring the transfer of information. Our findings again show that face-to-face interactions increase the accuracy, clarity and comprehensibility of complex medical information, thereby enabling relatives to participate in the care process. This parallels Molter’s [27] initial observation that the primary need of family members is to obtain information about the condition of their loved ones. On the other hand, although sources of dissatisfaction among families (inconsistent information, unclear roles, etc.) are known in the literature, this study has revealed in depth the significant impact of previous hospital experiences and negative preconceptions (e.g., ‘*Intensive care is not good*,’ ‘*Patients are treated badly’*) on the perception of the current situation and communication.

The analysis of these transcripts confirms that physician-provided information significantly affects patients’ relatives across multiple dimensions: emotionally, cognitively, and behaviorally. The content, frequency, and manner of information delivery all contribute to family members’ ability to cope with illness (e.g., “*I’ve been looking after him for 25 years. I get by on my husband’s pension. My husband isn’t retired, but I receive a care allowance and a disability allowance. We also get help from the local council, and our neighbours help out too. It’s very difficult, of course, I can’t meet every need”*.), make informed decisions, and develop appropriate expectations.

While the restriction of patient communication with relatives solely to telephone updates and the prohibition of physical visits may be justifiable on the grounds of infection control and safety, it generates significant ethical and organizational issues. In terms of autonomy, patients’ participation in decision-making processes through their families is impeded; furthermore, the inability of family members to observe the patient’s condition firsthand weakens the principle of informed consent and erodes the trust relationship with the medical team [37,38,39]. In terms of equity, the implemented restrictions have the potential to disproportionately affect certain groups. Specifically, families with limited digital access or a language barrier may struggle to comprehend information provided via telephone. In cultures where physical contact and the act of seeing the patient are considered culturally critical, such prohibitions can lead to ethical tension and cultural insensitivity [40,41].

Consequently, the principle of organizational justice necessitates a balanced approach between infection control and ensuring equitable access to emotional and informational support. Otherwise, hospital policies may undermine families’ sense of trust and belonging, leading to a perception of injustice [42]. When examining the psychological consequences of restrictive practices, one observes emotional responses such as patients experiencing feelings of loneliness and anxiety, while families exhibit intense anxiety and helplessness. This emotional strain also creates significant emotional exhaustion among healthcare professionals [43,44].

Therefore, mitigating the emotional and ethical impacts through technological and structural solutions such as video conferencing, specialized family communication units, and proactive transparent information delivery should be a fundamental strategy [39].

These findings emphasize the importance of thoughtful, consistent communication strategies that acknowledge both the informational and emotional needs of patients’ relatives. Healthcare professionals should recognize that their communication serves not only an informative function but also has profound effects on family members’ psychological well-being and participation in the healthcare process.

## 5. Conclusions

The main findings of this study highlight the urgency of structural and behavioural changes aimed at improving communication with patients’ relatives for healthcare systems and ICU teams. Prioritizing Face-to-Face Communication, Planning institutional policies and training that support effective face-to-face dialogue with patients’ relatives, adopting a balanced approach to ensure equitable access to emotional and informational support, and the implementation of technological and structural solutions such as video conferencing, dedicated family communication units, and proactive transparent information sharing to mitigate the emotional and ethical impacts of restrictive practices.

The methodological limitations of the study and the existing gaps in the literature suggest important fields for future research. Previous hospital experiences have been observed to have significant effects on families’ perception of the current situation and their level of engagement in communication. Future research should examine in detail how negative preconceptions and distrust shape current perceptions of communication and which communication strategies are most effective in changing these perceptions. It is also important to investigate whether the experiences of relatives of patients in third-level intensive care units in different hospitals (e.g., private hospitals or state hospitals in different regions) differ. In addition to this, future qualitative research, particularly when examining remote communication channels, may consider incorporating methodologies that enable the capture of non-verbal cues offered by face-to-face interaction (e.g., video call analysis).

## 6. Limitations

Our research was conducted with relatives of patients receiving treatment in a tertiary intensive care unit affiliated with a public hospital in Istanbul. This allowed us to understand the experiences of relatives of patients in intensive care at the institution in question with regard to the information-sharing process. However, it should be noted that the experiences of relatives of patients in third-level intensive care units at different hospitals may vary. Additionally, the use of only audio recordings in telephone interviews during the data collection process may have created a limitation in that the participants’ feelings and thoughts may not have been fully reflected. The exclusion of families of patients facing a significant risk of mortality and the selection of relatives of patients with a higher life expectancy led to a systematic selection bias in the sample. This situation is regarded as a methodological limitation of the study. It is recommended that future research, where feasible, should collect comparative data from families of patients who are at a high acute end-of-life risk.

## Figures and Tables

**Table 1 healthcare-13-03100-t001:** Descriptive statistics of the sociodemographic characteristics of ICU patients’ family members.

Variables	Characteristics	Frequency	Percentage %
Gender	Male	11	47.83
	Female	12	52.17
Age (years)	18–29	3	13.04
	30–49	16	69.56
	50–69	4	17.39
Patient’s Relative	Son	10	43.48
	Daughter	7	30.43
	Grandchild	2	8.7
	Mother	1	4.35
	Father	1	4.35
	Sister/Brother	1	4.35
	Nephew	1	4.35

**Table 2 healthcare-13-03100-t002:** Typology of communication problems in the delivery of intensive care.

Major Themes	Subthemes
Face-to-face information gathering experiences	Primary Information Channel Detailed and Comprehensive Information Sharing Building Trust and Changing Perceptions Psychological Relief Necessity of Visits
Experiences of obtaining information by telephone	Access Barriers and Communication Gaps Adapted Communication Channel (Long-Term Hospitalization) Secondary or Supportive Role
Communication competence of health professionals	Clear and Structured Presentation of Medical Information Use of understandable language and adaptation to the recipient Transparency and Expectation Management Proactive Communication and Accessibility Clear Instructions and Determining Next Steps Recognizing and Addressing Family Concerns
Adequacy of information	Detailed Explanations of Current Status and Treatment Plans Future Care Planning and Expectation Management Clear Instructions on Emergencies and Next Steps Channels and Frequency of Information Flow Perception of Family Knowledge Adequacy and Emotional Impact
Previous hospital experiences	Frequency and Scope of Experience Negative Preconceptions and Distrust Previous Care and Nursing Experience Long-Term Illness and Adaptation Continuity of Care

**Table 3 healthcare-13-03100-t003:** Face-to-Face Information Gathering Experiences.

Primary Information Channel	Patients’ relatives generally adopt daily face-to-face visits to the hospital as their primary method of gathering information. Some family members stated that they regularly visit the hospital to gather information face-to-face and are satisfied with this situation.
Detailed and Comprehensive Information Sharing	During face-to-face meetings, healthcare professionals provide detailed and technically comprehensive information on topics such as the patient’s current condition, treatment plans, potential outcomes, and laboratory results. This enables families to better understand the treatment process.
Building Trust and Changing Perceptions	Face-to-face interactions play an important role in strengthening families’ trust in healthcare professionals. For example, a family with negative preconceptions about intensive care stated that their perceptions changed thanks to face-to-face experiences and a continuous flow of information.
Psychological Relief	Receiving information face-to-face provides psychological relief for some relatives of patients and helps them manage uncertainty.
Necessity of Visits	In some cases, it has been noted that the flow of information is limited to face-to-face visits, which creates a physical obligation for family members who want to receive information to come to the hospital.

**Table 4 healthcare-13-03100-t004:** Experiences of Obtaining Information by Telephone.

Access Barriers and Communication Gaps	In one interview, it was clearly stated that family members’ attempts to call the hospital and speak with a nurse or doctor in the intensive care unit were blocked, and they were unable to obtain information by phone. This situation created a significant communication gap.
Adapted Communication Channel (Long-Term Hospitalization)	In contrast, in cases of long-term hospitalization (e.g., one year) and when family members have constraints such as work, daily updates by telephone have been found to be an established and critical channel of communication. In these cases, sufficient information can be provided by telephone on issues such as infection status, medication changes and ventilator status. The telephone has also been identified as a channel for obtaining information on weekends.
Secondary or Supportive Role	Information provided by telephone has generally served as a supportive or secondary channel in situations where visits are not possible, rather than replacing face-to-face communication.

**Table 5 healthcare-13-03100-t005:** Communication Competence of Health Professionals.

Clear and Structured Presentation of Medical Information	Healthcare professionals provide specific and detailed information about the patient’s condition, diagnoses (e.g., pneumonia, infection status), treatments (antibiotics, oxygen therapy, diuretics), and clinical improvement. They often present the treatment process in a chronological or logical order, including plans from the past to the present and future.
Use of understandable language and adaptation to the recipient	Although some medical terms may be used without explanation, professionals try to simplify complex medical information, use analogies, or ensure that they use language that is understandable to family members. They adapt their communication to the family member’s level of understanding and acknowledge the different communication needs of different family members (e.g., a son versus an elderly mother).
Transparency and Expectation Management	Professionals clearly communicate medical conditions, including the necessity of difficult procedures such as failed treatment attempts or tracheostomy. They manage expectations by discussing uncertainties in the process, avoiding definitive timelines, and reminding patients that their condition may be critical.
Proactive Communication and Accessibility	Healthcare professionals take the initiative to consult with other departments, make themselves available for questions, provide regular face-to-face updates to the family, and sometimes establish daily communication via telephone for distant family members.
Clear Instructions and Determining Next Steps	Clear guidance is provided regarding the family member’s potential transfers, visitation policies, and the outline of future care plans or treatment changes.
Recognizing and Addressing Family Concerns	Professionals anticipate potential complications, explain why certain approaches may be problematic in the long term, and monitor understanding.

**Table 6 healthcare-13-03100-t006:** Adequacy of Information.

Detailed Explanations of Current Status and Treatment Plans	Healthcare professionals generally provide detailed and regular information about the patient’s current clinical status, including lung condition, oxygen levels, heart function, infection status, and blood pressure. Additionally, information is provided about treatment plans and medication changes, such as antibiotics, oxygen therapy, diuretics, sedation, nutritional approach, and ventilation settings (e.g., “*The patient became hypoxemic, meaning their oxygen levels dropped, and they intermittently stopped breathing, their own efforts proving insufficient. That’s why, since they already have a tracheostomy, we cannot perform long-term intubation. Intubation after 14 days is something we don’t want, as it causes infection”*.).
Future Care Planning and Expectation Management	Professionals provide information about the patient’s future care pathway (e.g., transfer to a ward, need for tracheostomy, transition to palliative care) and potential procedures. They avoid false certainty by refraining from estimating the duration of treatment or by pointing out that recovery varies from patient to patient.(e.g., “*You may recall that we administered fluid loading a few weeks ago, after which the patient began to pass urine. Urine output is currently ongoing, but we may encounter a baseline kidney issue in the future. This means dialysis treatment may need to be reconsidered at some point. However, the patient is not currently undergoing active dialysis and has not received dialysis in the past. The antibiotic treatment will be completed next week. If you wish, we can discuss and plan with XX Hospital next Monday if there is availability”*.)
Clear Instructions on Emergencies and Next Steps	Clear instructions are provided on who to contact in emergencies, the need for family members to remain nearby in case of transfer, and visitation policies.
Channels and Frequency of Information Flow	Updates are typically provided face to face daily. In some cases, especially for family members who are far away, daily communication by phone is also provided. It is also noted that questions are asked of the patient’s family members regarding the adequacy and satisfaction with the information provided.
Perception of Family Knowledge Adequacy and Emotional Impact	Many family members are satisfied with the information provided and indicate that their basic information needs are being met. It has even been stated that receiving information provides psychological relief. (e.g., “*The doctor explained it so beautifully that all the questions I wanted to ask were already written down. I am very satisfied, extremely satisfied with everything, and particularly satisfied with the doctor. When explaining something in medical terms, she follows it up with the Turkish translation, and because she explains everything, I feel very at ease. Thank you very much”*.)

**Table 7 healthcare-13-03100-t007:** Previous Hospital Experiences.

Frequency and Scope of Experience	Most sources minimally address previous hospital experiences. However, in one interview, preconceptions about intensive care experiences were quite prominent.
Negative Preconceptions and Distrust	It was observed that families had negative preconceptions about intensive care (such as ‘intensive care is not good’ or ‘patients are treated poorly’) and that this influenced their initial treatment decisions (e.g., refusing admission to intensive care). In some cases, uncertainty in the initial diagnosis and inconsistencies among doctors were reported to have led to distrust. (e.g., “*At first, those around us said, ‘Intensive care isn’t good.’ They said, ‘They don’t look after the patients there; they beat them up.’ That idea became so ingrained in our minds that we didn’t send him to intensive care at first. Then, of course, when my brother’s condition worsened again and there was nothing else we could do, we brought him here”*.)
Previous Care and Nursing Experience	It has been stated that some family members have previous experience of care in a hospital environment and are even capable of performing basic nursing tasks such as administering IVs and managing oxygen (e.g., “*Our nurse, our head nurse, provides training for patients who will be going to palliative care or the ward, for example. Monitoring patients on ventilators and caring for patients in particular requires training”*.). Having a family member who is a nurse may influence the understanding of and expectations for hospital care. Experience of providing long-term care may also be relevant.
Long-Term Illness and Adaptation	It has been noted that some patients have been hospitalized for approximately one year and have undergone multiple intensive care admissions. This situation has led to a significant adaptation process, requiring the family to alter their living arrangements and develop established communication routines for obtaining information.(e.g., “*My brother has been in intensive care for six days now. Before that, he was in a regular ward (he was in the ward). We have been battling this illness for 2.5 to 3 years*”.)
Continuity of Care	References to previous medical interactions, such as a prior hip fracture, oncological treatment, or transfer from another facility, provide context for the current hospital stay and demonstrate continuity of care. Familiarity with hospital routines and visitation policies has also been observed.

## Data Availability

The data presented in this study are available on request from the corresponding author. The data are not publicly available due to corporate and personal confidentialities. The dataset of the study involves the deep interviews conducted with the real patients’ relatives who took care in a state hospital in Turkey.

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
