# Peer review of "Interpersonal Communication in Intensive Care Units: A Qualitative Study on Family Members’ Experiences in a Turkish Public Hospital"

_healthcare, 2025, doi:10.3390/healthcare13233100_

Round 1

Reviewer 1 Report

Comments and Suggestions for Authors

General Assessment:

The manuscript addresses a relevant and underexplored topic within healthcare communication and family engagement in intensive care settings. However, several critical issues limit its suitability for publication in its current form.

Introduction:

The theoretical framework presented in the introduction lacks coherence and structure. The rationale for integrating the concepts of communication, health literacy, family needs, and satisfaction is insufficiently developed. A clearer articulation of the interrelationship among these themes is necessary to justify the study’s scope and objectives.

Materials and Methods:

While the authors outline the general procedures for data collection and analysis, the sampling strategy remains ambiguous. Further clarification is needed regarding participant selection criteria and recruitment processes. Additionally, the manuscript would benefit from a more detailed contextual description of the two intensive care units (ICU 1 and ICU 2), which is essential for interpreting the findings.

Results:

The sociodemographic data of the participants are presented with clarity. However, the manuscript does not specify the clinical reasons for the patients’ hospitalisation, which could provide valuable context for understanding family members’ experiences. Although the thematic categories derived from the qualitative analysis are listed in tables, the absence of illustrative excerpts from the interviews weakens the depth and credibility of the qualitative findings.

Discussion:

The discussion section lacks engagement with existing literature. No references to prior studies or theoretical perspectives are provided, which undermines the scholarly value of the analysis. The authors should integrate relevant citations to situate their findings within the broader academic discourse.

Conclusion:

The manuscript does not include a distinct conclusion section. A summarising subchapter that synthesises the key findings and outlines implications for practice and future research is recommended.

Recommendation:

Given the aforementioned limitations in theoretical grounding, methodological transparency, data presentation, and scholarly contextualisation, I regret to recommend rejection of the manuscript in its current form.

Author Response

Comments 1: 

Introduction:

The theoretical framework presented in the introduction lacks coherence and structure. The rationale for integrating the concepts of communication, health literacy, family needs, and satisfaction is insufficiently developed. A clearer articulation of the interrelationship among these themes is necessary to justify the study’s scope and objectives.

Response 1: Dear Reviewer, thank you very much for your constructive, valuable and informative guidance.

After reading your comments, we reread the text and observed that we had not sufficiently defined the concept of health literacy, had not linked it to the research findings, and had not even designed the study accordingly. Therefore, we removed the health literacy concepts from the article's title and text, along with the places where we had linked them, and reformulated them in line with the communication theories we included in our study.

Comments 2: 

Materials and Methods:

While the authors outline the general procedures for data collection and analysis, the sampling strategy remains ambiguous. Further clarification is needed regarding participant selection criteria and recruitment processes. Additionally, the manuscript would benefit from a more detailed contextual description of the two intensive care units (ICU 1 and ICU 2), which is essential for interpreting the findings.

Response 2: We have elaborated the methodology section for greater clarity and transparency. Thus, we have specified study's approach and epistemological stance as constructivist, or interpretivist.

We have clearly described how participants were selected, what inclusion and exclusion criteria were applied beyond the 72-hour hospitalization threshold.

We have explained how saturation was determined, including the criteria used and the number of interviews conducted before no new codes emerged.

We have provided details about the interviewer's background and their relationship with the participants. 

Comments 3: 

Results:

The sociodemographic data of the participants are presented with clarity. However, the manuscript does not specify the clinical reasons for the patients’ hospitalisation, which could provide valuable context for understanding family members’ experiences. Although the thematic categories derived from the qualitative analysis are listed in tables, the absence of illustrative excerpts from the interviews weakens the depth and credibility of the qualitative findings.

Response 3: We have specified the clinical reasons for the patients' hospitalisation such as: 

The primary reasons for ICU admission of the selected patients were pneumonia(n=5), respiratory insufficiency due to chronic obstructive lung disease(n=4), sepsis(n=3), postoperative hemodynamic-respiratory problems(n=3), gastrointestinal hemorrhage(n=1), intracranial bleeding-stroke(n=2) , acute heart failure (n=1),  renal failure(n=2), acute pancreatitis(n=1), diabetic coma(n=1). The patients under the severe risk of death were excluded from the study (in shock state, terminal malignancy, vegetative state etc.)

Comments 4: 

Discussion:

The discussion section lacks engagement with existing literature. No references to prior studies or theoretical perspectives are provided, which undermines the scholarly value of the analysis. The authors should integrate relevant citations to situate their findings within the broader academic discourse.

Conclusion:

The manuscript does not include a distinct conclusion section. A summarising subchapter that synthesises the key findings and outlines implications for practice and future research is recommended.

Response 4: The discussion and conclusion have been rewritten in relation to the communication theories we have cited in the manuscript as Uncertainty Management Theory, Information Management, Relationship-Centered Care and Health Belief Model. We have also included some more quotations examples from our interviews in discussion section. We have tried to show how communication mode affects emotional well-being, trust, and comprehension of the patients' families. We have cited prior studies to reflect the theoretical and methodological perspectives of the study.

We have added some cultural context to be able to reflect on Turkish ICU culture and hierarchical communication norms that may shape family interactions differently than in Western contexts. This could be inspiring and useful for future work in this field.

The sources cited in our study generally supported the importance of communication in the ICU and emphasised the limited number of qualitative/theoretical studies in this field. However, following your comments and suggestions, we also focused on references that shed light on the complexities, ethical conflicts and limitations in the application of interpersonal communication, thereby bringing a critical perspective to the academic examination of the subject. For examples; the studies focusing on how visitation restrictions and practices such as providing information only by telephone hinder patients' families' participation in decision-making processes, undermine the principle of informed consent, and ultimately erode trust in the medical team:
    â—¦ Dudgale, L.S. et al. (2023): Ethical Guidance on Family Care, Support, and Visitation in Hospitals and Care Homes.
    â—¦ Azoulay, E. et al. (2020): Family in the ICU: Decision-Making Capacity and Support.
    â—¦ Davidson, J.E. et al. (2017): Family-centred Care Guidelines in Neonatal, Paediatric and Adult ICUs.

and the studies suggesting that restrictions may lead to ethical tensions and cultural insensitivity in situations where physical contact or the act of seeing the patient is culturally critical:
â—¦ Honarmand, K. and Mehta, S. (2021): The Consequences of Visitor Restriction Policies in ICUs During the COVID-19 Pandemic.
    â—¦ Campbell, L. and Morley, G. (2023): The Ethical Complexity of Restricting Visitors During the COVID-19 Pandemic.

A conclusion section summarising the main findings and outlining implications for practice and future research has been written and added to the main text.

We have highlighted these in yellow within the text. Please see the attachment.

Best Regards.

Reviewer 2 Report

Comments and Suggestions for Authors

Dear authors,

I read your manuscript titled "Interpersonal Communication and Health Literacy in Intensive Care Units: A Qualitative Study on Family Members' Experiences in a Turkish Public Hospital" with great interest. This engaging and contextually significant study explores the interplay between communication, family satisfaction, and health literacy in intensive care units (ICUs). This is an area that is often overlooked in the critical care literature, especially in low-resource, high-demand settings such as public hospitals.

Your work addresses a genuine gap in the literature, employs a qualitative methodology, and provides valuable insights into how different modes of communication (face-to-face vs. telephone) impact relatives' emotional and informational experiences. However, the study could benefit from more extensive methodological clarification, tighter structure, and a more critical discussion of its implications and limitations. This is the focus of my review.

Overall appraisal

Strengths

1- Timely topic with high clinical and ethical relevance.

2- Focus on family–clinician communication under resource constraints, which adds real-world applicability.

3- Precise thematic categorization and comprehensive qualitative data presentation.

4- Inclusion of both communication theory and family satisfaction perspectives.

Main weaknesses

1-Methodological description lacks precision and transparency.

2- The data analysis process is underdeveloped and partially inconsistent with qualitative rigor.

3- The results section overlaps with the discussion and lacks a deeper interpretive synthesis.

4-Theoretical grounding (communication and health literacy frameworks) is mentioned but not adequately operationalized.

5-Language and structure require refinement to achieve clarity and an academic tone.

Major comments

1. Clarity and transparency of methodology

While the study is presented as a qualitative descriptive design that utilizes semi-structured interviews and content analysis, its description lacks clarity and contains repetitive elements. To ensure reproducibility and credibility, the following aspects need to be addressed:

1. Specify the Epistemological Stance: Indicate whether the approach is phenomenological, constructivist, or interpretivist.

2. Participant Selection: Clearly describe how participants were selected. Was purposive or convenience sampling used? What inclusion and exclusion criteria were applied beyond the 72-hour hospitalization threshold?

3. Saturation Determination: Explain how saturation was determined, including the criteria used and the number of interviews conducted before no new codes emerged.

4. Interviewer Background: Provide details about the interviewer's background and their relationship with the participants. Include the steps taken to minimize bias, such as a reflexivity statement.

5. Coding Process in Atlas.ti: Describe the coding process, including the number of coders involved and whether intercoder reliability was assessed. Was the codebook developed iteratively?

6. Content vs. Thematic Analysis: Clearly distinguish between content analysis and thematic analysis, as both are referenced. Choose one approach and justify it (e.g., using inductive thematic analysis as described by Braun & Clarke).

Without addressing these elements, the study's qualitative rigor may be insufficient for publication.

2. Study design and data context

The comparison between the two ICUs (ICU-1: telephone-only communication, ICU-2: face-to-face interaction with visitation) is intriguing but has a quasi-experimental nature. However, the manuscript presents this comparison descriptively.

To enhance clarity and interpretability, consider the following suggestions:

1. Explicitly frame the study design as a comparative, descriptive, qualitative study.

2. Clarify whether the differences in protocols were intentional for the study or part of existing practices.

3. Discuss the potential organizational and ethical implications of restricting visits in ICU-1, including autonomy, equity, and psychological impact.

4. Address the possibility of selection bias—were families in ICU-2 systematically different in terms of education, socioeconomic status, or prior experiences?

3. Analysis and presentation of findings

The tables (3–7) provide valuable information but are often overly detailed and occasionally repetitive. To enhance the analytical depth, consider the following suggestions:

1. Condense the tables by merging overlapping subthemes, such as "tailored and Comprehensive Information Sharing" with "adequacy of Information." 2. Focus on highlighting contrasts between communication modes instead of reiterating neutral descriptions.

3. Include verbatim quotations from participants to support key points; qualitative research relies on direct evidence to convey authenticity and emotional nuance.

4. Clearly explain how health literacy emerged as a theme; currently, the results address communication content and emotional responses, but do not address literacy specifically.

5. The lack of quotations makes the analysis feel more like a report than an interpretive synthesis.

4. Integration of theory

The introduction references communication theory, uncertainty management, and shared decision-making, but the analysis does not systematically engage with these frameworks.

Strengthen this by:

  • Explicitly linking themes (e.g., "rust, " "motivational relief, "" clarity of language") to uncertainty reduction theory or information management theory.
  • Defining "health literacy" and clarifying how it was assessed — the current study does not actually measure literacy but infers it from communication comprehension. Consider reframing to "received communication clarity and informational understanding." Discuss power dynamics in clinician–family relationships and how communication shapes perceived agency and satisfaction.

Anchoring findings in a theoretical lens would significantly improve depth and coherence.

5. Discussion and interpretation

The discussion restates results rather than critically analyzing them. Consider restructuring as follows:

  1. Interpretive synthesis: Show how communication mode affects emotional well-being, trust, and comprehension.
  2. Mechanisms: Explain why face-to-face interactions succeed — not just that they do. For instance, emphasize non-verbal cues, empathy, and immediacy.
  3. Comparative insight: Situate your findings within the existing literature (e.g., Haave et al., 2021; Ferrando et al., 2019; Stricker et al., 2009). Highlight consistencies and novel contributions.
  4. Policy and practice implications: Suggest concrete interventions such as structured family meetings, designated communication officers, or hybrid communication models (e.g., scheduled video updates).
  5. Cultural context: Reflect on Turkish ICU culture and hierarchical communication norms that may shape family interactions differently than in Western contexts.

This will shift the discussion from descriptive to analytical, making it more compelling.

6. Limitations and ethical considerations

The limitations section is adequate, but should expand on:

  • Transferability — findings from one public hospital may not generalize to private or rural facilities.
  • Researcher positionality — were the interviewers clinicians? How might that have influenced participants ' candor?
  • Language and translation — interviews conducted in Turkish: who translated, and how was meaning preserved?

Acknowledging these aspects demonstrates reflexivity and enhances credibility.

Minor comments

  1. Title and abstract:
    • The title is informative but could be shorter ("Interpersonal Communication and Family Experiences in Turkish Intensive Care Units").
    • The abstract should include the sample size (n = 23), method (semi-structured interviews), analytic approach (content or thematic analysis), and key themes.
  2. Writing and language:
    • The manuscript would benefit from professional English editing to improve fluency and syntax (e.g., "uth forward" → ""ut forward"" ""rofessionnals" → ""rofessionals"".
    • Avoid colloquial expressions and redundant phrasing.
  3. Tables and figures:
    • Simplify table layout; use shorter phrases instead of complete sentences.
    • Include one summary figure that maps communication modes to outcomes (trust, comprehension, emotional relief).
  4. References:
    • The literature review is broad but outdated in some areas (e.g., Molter, 1979). Add recent family communication studies (published after 2020) for stronger grounding.
    • Check citation formatting for consistency (APA or MDPI style).

Encouragement

Despite the methodological limitations, this is a valuable and heartfelt study that captures the human dimension of intensive care — how families make sense of crisis through communication. The qualitative approach is appropriate, and the insights have practical implications for improving ICU-family relations, especially in public hospitals with limited resources.

If the authors revise the methodology to enhance transparency, streamline the results, and strengthen the theoretical framing, this paper could make a meaningful contribution to the literature on patient- and family-centered critical care communication.

Recommendation

Major revision.

Revisions should focus on:

  1. Methodological transparency (sampling, coding, analysis).
  2. Integration of Communication and Health Literacy Theory.
  3. Inclusion of direct quotations and deeper interpretive analysis.
  4. More transparent structure and refined language.

Once improved, the manuscript has strong potential for publication, as it addresses a real clinical and ethical need — building trust and understanding through human connection in the ICU.

Comments on the Quality of English Language

See authors comment.

Author Response

Comments 1:

  1. Clarity and transparency of methodology

While the study is presented as a qualitative descriptive design that utilizes semi-structured interviews and content analysis, its description lacks clarity and contains repetitive elements. To ensure reproducibility and credibility, the following aspects need to be addressed:

  1. Specify the Epistemological Stance: Indicate whether the approach is phenomenological, constructivist, or interpretivist.
  2. Participant Selection: Clearly describe how participants were selected. Was purposive or convenience sampling used? What inclusion and exclusion criteria were applied beyond the 72-hour hospitalization threshold?
  3. Saturation Determination: Explain how saturation was determined, including the criteria used and the number of interviews conducted before no new codes emerged.
  4. Interviewer Background: Provide details about the interviewer's background and their relationship with the participants. Include the steps taken to minimize bias, such as a reflexivity statement.
  5. Coding Process in Atlas.ti: Describe the coding process, including the number of coders involved and whether intercoder reliability was assessed. Was the codebook developed iteratively?
  6. Content vs. Thematic Analysis: Clearly distinguish between content analysis and thematic analysis, as both are referenced. Choose one approach and justify it (e.g., using inductive thematic analysis as described by Braun & Clarke).

Without addressing these elements, the study's qualitative rigor may be insufficient for publication.

Response 1: Dear Reviewer, thank you very much for your constructive, valuable and informative guidance. 

We have elaborated the methodology section for greater clarity and transparency. Thus, we have specified study's approach and epistemological stance as constructivist, or interpretivist.
We have clearly described how participants were selected, what inclusion and exclusion criteria were applied beyond the 72-hour hospitalization threshold.
We have explained how saturation was determined, including the criteria used and the number of interviews conducted before no new codes emerged.
We have provided details about the interviewer's background and their relationship with the participants. The coding process has been detailed in "data collection" section. After reading your comments, we reread the text and realised that proceeding with a thematic analysis-focused research method would be more appropriate. Thank you again. 

Comments 2: 

2. Study design and data context

The comparison between the two ICUs (ICU-1: telephone-only communication, ICU-2: face-to-face interaction with visitation) is intriguing but has a quasi-experimental nature. However, the manuscript presents this comparison descriptively.

To enhance clarity and interpretability, consider the following suggestions:

1. Explicitly frame the study design as a comparative, descriptive, qualitative study.

2. Clarify whether the differences in protocols were intentional for the study or part of existing practices.

3. Discuss the potential organizational and ethical implications of restricting visits in ICU-1, including autonomy, equity, and psychological impact.

4. Address the possibility of selection bias—were families in ICU-2 systematically different in terms of education, socioeconomic status, or prior experiences?

Response 2: 

The differences in protocols between ICU 1 and 2 have been detailed by our author, who is the responsible intensive care physician.

The potential organisational and ethical implications of restricting visits in ICU-1, including autonomy, equity, and psychological impact, have been detailed in both the methodology and discussion sections with the contribution of our author, who is a medical ethics expert. 

It is noted in a footnote in the study population section that families in ICU-2 did not show any systematic differences in terms of education, socio-economic status or previous experiences.

Comments 3: 

3. Analysis and presentation of findings

The tables (3–7) provide valuable information but are often overly detailed and occasionally repetitive. To enhance the analytical depth, consider the following suggestions:

1. Condense the tables by merging overlapping subthemes, such as "tailored and Comprehensive Information Sharing" with "adequacy of Information." 2. Focus on highlighting contrasts between communication modes instead of reiterating neutral descriptions.

3. Include verbatim quotations from participants to support key points; qualitative research relies on direct evidence to convey authenticity and emotional nuance.

4. Clearly explain how health literacy emerged as a theme; currently, the results address communication content and emotional responses, but do not address literacy specifically.

5. The lack of quotations makes the analysis feel more like a report than an interpretive synthesis.

Response 3:

The results have been rewritten in relation to the communication theories we have cited in the literature and the main and sub-themes we have selected for thematic analysis. We have also included some more quotations examples from our interviews in results section. 

After reading your comments, we reread the text and observed that we had not sufficiently defined the concept of health literacy, had not linked it to the research findings, and had not even designed the study accordingly. Therefore, we removed the health literacy concepts from the article's title and text, along with the places where we had linked them, and reformulated them in line with the communication theories we included in our study.

Comments 4: 

4. Integration of theory

The introduction references communication theory, uncertainty management, and shared decision-making, but the analysis does not systematically engage with these frameworks.

Strengthen this by:

  • Explicitly linking themes (e.g., "rust, " "motivational relief, "" clarity of language") to uncertainty reduction theory or information management theory.
  • Defining "health literacy" and clarifying how it was assessed — the current study does not actually measure literacy but infers it from communication comprehension. Consider reframing to "received communication clarity and informational understanding." Discuss power dynamics in clinician–family relationships and how communication shapes perceived agency and satisfaction.

Anchoring findings in a theoretical lens would significantly improve depth and coherence.

Comments 5:

5. Discussion and interpretation

The discussion restates results rather than critically analyzing them. Consider restructuring as follows:

  1. Interpretive synthesis: Show how communication mode affects emotional well-being, trust, and comprehension.
  2. Mechanisms: Explain why face-to-face interactions succeed — not just that they do. For instance, emphasize non-verbal cues, empathy, and immediacy.
  3. Comparative insight: Situate your findings within the existing literature (e.g., Haave et al., 2021; Ferrando et al., 2019; Stricker et al., 2009). Highlight consistencies and novel contributions.
  4. Policy and practice implications: Suggest concrete interventions such as structured family meetings, designated communication officers, or hybrid communication models (e.g., scheduled video updates).
  5. Cultural context: Reflect on Turkish ICU culture and hierarchical communication norms that may shape family interactions differently than in Western contexts.

Response 4-5:

The discussion and conclusion have been rewritten in relation to the communication theories we have cited in the manuscript as Uncertainty Management Theory, Information Management, Relationship-Centered Care and Health Belief Model. We have also included some more quotations examples from our interviews in discussion section. We have tried to show how communication mode affects emotional well-being, trust, and comprehension of the patients' families. We have cited prior studies to reflect the theoretical and methodological perspectives of the study.

We have added some cultural context to be able to reflect on Turkish ICU culture and hierarchical communication norms that may shape family interactions differently than in Western contexts. This could be inspiring and useful for future work in this field.

We have situated our findings within the existing literature (e.g., Haave et al., 2021; Ferrando et al., 2019; Stricker et al., 2009) and highlighted consistencies and novel contributions.

The sources cited in our study generally supported the importance of communication in the ICU and emphasised the limited number of qualitative/theoretical studies in this field. However, following your comments and suggestions, we also focused on references that shed light on the complexities, ethical conflicts and limitations in the application of interpersonal communication, thereby bringing a critical perspective to the academic examination of the subject. For examples; the studies focusing on how visitation restrictions and practices such as providing information only by telephone hinder patients' families' participation in decision-making processes, undermine the principle of informed consent, and ultimately erode trust in the medical team:
    â—¦ Dudgale, L.S. et al. (2023): Ethical Guidance on Family Care, Support, and Visitation in Hospitals and Care Homes.
    â—¦ Azoulay, E. et al. (2020): Family in the ICU: Decision-Making Capacity and Support.
    â—¦ Davidson, J.E. et al. (2017): Family-centred Care Guidelines in Neonatal, Paediatric and Adult ICUs.
and the studies suggesting that restrictions may lead to ethical tensions and cultural insensitivity in situations where physical contact or the act of seeing the patient is culturally critical:
â—¦ Honarmand, K. and Mehta, S. (2021): The Consequences of Visitor Restriction Policies in ICUs During the COVID-19 Pandemic.
    â—¦ Campbell, L. and Morley, G. (2023): The Ethical Complexity of Restricting Visitors During the COVID-19 Pandemic.

A conclusion section summarising the main findings and outlining implications for practice and future research has been written and added to the main text.

We have highlighted these in yellow within the text. Please see the attachment.

Best Regards.

Reviewer 3 Report

Comments and Suggestions for Authors

I find the research very relevant, especially the key theme: understanding how families of patients admitted to the ICU feel.

I overlooked that, during the presentation of the results, the authors included family members' statements expressing each of the study's themes and subthemes. Therefore, I believe they should add these statements to the results section, although they did include some in the discussion section.

Author Response

Comment 1: I overlooked that, during the presentation of the results, the authors included family members' statements expressing each of the study's themes and subthemes. Therefore, I believe they should add these statements to the results section, although they did include some in the discussion section.

Response 1: Dear Reviewer, thank you very much for your constructive, valuable and informative guidance. 

The results have been rewritten in relation to the communication theories we have cited in the literature and the main and sub-themes we have selected for thematic analysis. We have also included some more quotations examples from our interviews in results section. We have highlighted these in yellow within the text. Please see the attachment.

Best Regards.

Round 2

Reviewer 1 Report

Comments and Suggestions for Authors

The authors of the article are to be congratulated for the review conducted. I consider that the content gives relevance to the topic and to the methodological construct. They are thus able to highlight the results achieved, discuss them, and present the contributions to this thematic area. It can therefore be accepted in its current form.

Author Response

Comments 1: The authors of the article are to be congratulated for the review conducted. I consider that the content gives relevance to the topic and to the methodological construct. They are thus able to highlight the results achieved, discuss them, and present the contributions to this thematic area. It can therefore be accepted in its current form.

Response 1: Thank you again for your constructive and supportive attitude and valuable opinions.

Reviewer 2 Report

Comments and Suggestions for Authors

The authors have meaningfully expanded the Methods and now explicitly state a constructivist/interpretivist stance; they describe the two settings and protocols, participant selection, and their use of Atlas.ti; they present a clearer theme structure and include more verbatim quotes; they align the framing with Uncertainty Management Theory, Information Management, Relationship-Centered Care, and Health Belief Model; they add ethics approval and consent details and clarify that interviews were audio-recorded. These are welcome steps.   Major comments 2.1. Methodological coherence: approach and analytic technique
The manuscript still mixes “content analysis” and “thematic analysis,” and even includes a word-cloud step, which is unnecessary for a small, interview-based dataset. Please commit to one primary analytic approach and justify it (e.g., reflexive thematic analysis, as outlined by Braun & Clarke, with clear phases and operationalization of each; or a conventional qualitative content analysis, as described by Hsieh & Shannon). Currently, the execution is described as hybrid without an explicit rationale. Map your current steps to the chosen framework in a tight paragraph. 2.2. Sampling strategy: contradictory labels and selection boundaries
You describe purposive plus convenience sampling in the Study Population section and then refer to it as “homogeneous sampling” in Data Collection. These labels have different meanings. Please choose and defend the single strategy actually used. If purposive, specify the purposive criteria; if homogeneous, define the shared characteristics and why they matter to the research question. Also, you explicitly excluded families of patients “under severe risk of death,” selecting relatives of patients “with higher life expectancy” so they could “better evaluate the communication service.” This introduces a systematic perspective bias. It must be acknowledged upfront as a limitation and, if possible, contrasted with at least a small number of “high-acuity end-of-life” cases in future work. 2.3. Comparative design and timing of protocol changes
The two-ICU contrast remains one of your most interesting aspects, but the manuscript states that the two different communication/visitation protocols “were started to be performed two weeks before the study was begun.” This implies a quasi-experimental organizational change enacted just prior to data collection, with possible novelty and Hawthorne effects, contamination, and staff learning curves. Please clarify whether these protocols were pre-existing institutional practices or were specifically instituted for the study. If the latter, acknowledge the potential threats to credibility/transferability and discuss the mitigation strategies. Provide a simple table with participant characteristics by ICU (ICU-1 vs ICU-2) to support your assertion that the family groups were similar; at a minimum, show age, relationship to patient, prior ICU experience, and education by ICU. Currently, the text asserts “no differences” without providing evidence. 2.4. Rigor and trustworthiness: reflexivity, coding, and reliability
You add an interviewer footnote and report very high Cohen’s kappa values (κ=.91–1.00) after “two rounds of training,” with one coder coding all interviews and a second coding 20% (n=5). Please expand rigor reporting along accepted qualitative criteria:

  • Reflexivity: a dedicated paragraph on researchers’ positionality, assumptions, prior ICU experience, and how reflexive memoing influenced interpretation.
  • Coding process: how the codebook was developed (inductively/deductively), how disagreements were handled beyond kappa, and how themes were generated from codes.
  • Reliability vs reflexivity: explain why you used kappa (a quantitative index) in an interpretivist design and how you balanced reliability metrics with reflexive thematic development. Provide an audit trail figure (from raw data to codes, to categories, to themes). If possible, include a codebook excerpt in the Supplementary Material.

2.5. Saturation claim
You define saturation as the point “no new thematic element emerged” and assert it was reached at n=23. Please make this claim more transparent: describe how you tracked code/theme accrual over interviews (e.g., a brief saturation grid) and at which interview you first observed thematic stability; otherwise, the statement risks sounding tautological. 2.6. Use of theory in analysis rather than post-hoc in discussion
You now anchor the discussion in several frameworks, which is good. However, the analysis section still reads largely descriptive. Strengthen the analytic narrative by explicitly showing where specific codes/subthemes instantiate each theory in the Results (e.g., how a quote evidences uncertainty reduction through information structure; how relational behaviors support Relationship-Centered Care; how perceived severity/benefit cues map to Health Belief Model). A compact table mapping themes → theoretical concepts → exemplar quotes would materially improve coherence. 2.7. Quotations: thick description and negative cases
You have added quotes, but a more purposeful use is needed. For each major theme contrasting face-to-face interactions with telephone interactions, include at least one confirming and one disconfirming/nuanced quote (“negative case analysis”) to avoid a one-sided story and to demonstrate analytic depth. Where possible, indicate the ICU and role (e.g., daughter, son) for each quote, preserving anonymity. Also add a sentence on translation procedures (Turkish→English), who translated, and how meaning was preserved (back-translation or team consensus). 2.8. Claims of equivalence and generalization
Statements such as “communication competence” being “generally strong” and that medical jargon “did not impede comprehension” are broad. Support these with systematic evidence (e.g., proportion of interviews where jargon was flagged as confusing) or temper them. Similarly, the policy implications should be framed as suggestions consistent with your data and context, not generalized prescriptions. Consider turning the policy bullets into an ICU practice checklist box explicitly labeled as practice suggestions, not guidelines. 2.9. Ethics of restricted visitation: integrate with results, not only discussion
The ethical analysis is strong, but reads as a parallel essay. Bring at least one analytic thread into Results where participants’ words explicitly speak to autonomy/equity/trust under restrictions; then expand in Discussion with the literature you cite. This will align claims more tightly to data. 2.10. Reporting standards and transparency
Please attach a completed COREQ checklist and consider providing, in the Supplementary Materials, the semistructured interview guide, a codebook excerpt, a brief saturation grid, and a theme map. Specify the software versions already used (e.g., SPSS, Atlas.ti). The current Data Availability Statement can remain, but consider offering de-identified quote tables if full transcripts cannot be shared.

  1. Minor comments (but please address all)

a) Tables: Table 1 is referenced, but in the PDF appears as a blank layout before the variable listing; ensure all tables render correctly with clear headings, denominators, and consistent formatting. Standardize decimal points (dot, not comma) and tidy typography.
b) Language/typos: correct “puth forward,” “professionnals,” “delvery,” and similar minor errors; standardize terms (face-to-face, telephone). Replace metaphor paragraphs (e.g., the “map” analogy) with analytic prose or move to a brief rhetorical flourish in the Discussion if you wish to keep it.
c) Setting detail: the occupancy rate “95–110%,” nurse:patient ratio, and mortality band are informative but could be condensed; ensure these context markers are used to interpret communication logistics (e.g., feasibility of daily face-to-face updates under high occupancy).
d) Claims of “no differences between ICU-1 and ICU-2 families”: support with a small comparative table, or remove the claim.
e) Clarify whether any participants received both modalities during the stay (face-to-face some days, telephone others), and how you treated those experiences analytically.
f) Add a brief statement on data security and anonymization procedures for audio files and transcripts.

Comments on the Quality of English Language

See comment.

Author Response

Response 1: Thank you again for your constructive and supportive attitude and valuable opinions. 

We have uploaded all your suggested additions and corrections as separate files titled "revised manuscript V2" and "supplementary materials." We've highlighted our new edits in green within the "revised manuscript V2" file. Please see the attachment. 
